# Composite Nanoarchitectonics with CoS_2_ Nanoparticles Embedded in Graphene Sheets for an Anode for Lithium-Ion Batteries

**DOI:** 10.3390/nano12040724

**Published:** 2022-02-21

**Authors:** Tongjun Li, Hongyu Dong, Zhenpu Shi, Hongyun Yue, Yanhong Yin, Xiangnan Li, Huishuang Zhang, Xianli Wu, Baojun Li, Shuting Yang

**Affiliations:** 1School of Physics, Henan Normal University, Xinxiang 453007, China; tongjunxiaoyu@163.com (T.L.); zhenpushi@foxmail.com (Z.S.); lxn506@163.com (X.L.); zhanghuishuang0000@163.com (H.Z.); 2School of Chemistry and Chemical Engineering, Henan Normal University, Xinxiang 453007, China; yuehongyun@foxmail.com (H.Y.); yinyh@163.com (Y.Y.); 3National and Local Joint Engineering Laboratory of Motive Power and Key Materials, Henan Normal University, Xinxiang 453007, China; 4Collaborative Innovation Center of Henan Province for Motive Power and Key Materials, Henan Normal University, Xinxiang 453007, China; 5College of Chemistry, Zhengzhou University, Zhengzhou 453000, China; wuxianli@zzu.edu.cn (X.W.); lbjfcl@zzu.edu.cn (B.L.)

**Keywords:** CoS_2_@rGO, in-situ XRD, lithium-ion battery, anode

## Abstract

Cobalt sulfides are attractive as intriguing candidates for anodes in Lithium-ion batteries (LIBs) due to their unique chemical and physical properties. In this work, CoS_2_@rGO (CSG) was synthesized by a hydrothermal method. TEM showed that CoS_2_ nanoparticles have an average particle size of 40 nm and were uniformly embedded in the surface of rGO. The battery electrode was prepared with this nanocomposite material and the charge and discharge performance was tested. The specific capacity, rate, and cycle stability of the battery were systematically analyzed. In situ XRD was used to study the electrochemical transformation mechanism of the material. The test results shows that the first discharge specific capacity of this nanocomposite reaches 1176.1 mAhg^−1^, and the specific capacity retention rate is 61.5% after 100 cycles, which was 47.5% higher than that of the pure CoS_2_ nanomaterial. When the rate changes from 5.0 C to 0.2 C, the charge-discharge specific capacity of the nanocomposite material can almost be restored to the initial capacity. The above results show that the CSG nanocomposites as a lithium-ion battery anode electrode has a high reversible specific capacity, better rate performance, and excellent cycle performance.

## 1. Introduction

Lithium-ion batteries (LIBs) have attracted wide attention due to their high energy density, good cycling stability, and no memory effect [1,2]. Since their introduction in 1991, graphite has been the dominant commercial anode materials for lithium-ion batteries, due primarily to the low theoretical specific capacity of graphite (372 mAhg^−1^). However, they cannot meet the requirements of high energy density lithium-ion batteries. The search for anode materials with higher power density and energy density performance is a concern of modern scientific researchers [3]. In the research of lithium-ion batteries, sulfide anodes have attracted much attention due to their high theoretical capacity. Compared with oxides, sulfides have lower electronegativity, a more flexible structure, and demonstrate better performance. Compared with other anode materials, the morphology design of sulfides is easier and more controllable with higher specific capacity [3,4]. However, sulfides as lithium-ion electrode materials have disadvantages such as poor conductivity and poor cycle stability. Therefore, improving the properties of sulfides has become a current research focus. So that the transition metal sulfide as the anode materials of lithium-ion batteries is severely limited [5]. Graphene, as a 2D nanosheet, has important applications in energy materials and micro-nano processing due to its excellent optical, electrical and mechanical properties. Reduced graphene oxide (rGO) is easy to prepare and has the advantages of extremely thin interlayer spacing (about 0.36 nm), large in-plane size, high electronic conductivity, and good chemical stability. It is widely used as an auxiliary material for lithium battery materials [6]. The microstructure and chemical stability of the electrode can buffer the volume change of the active substance in the charging and discharging process to a certain extent. Many research results show that rGO plays an active role in electrode composites [7,8,9]. Notably, rGO also prevents the aggregation and agglomeration of inorganic nanoparticles. CoS_2_ has high theoretical specific capacity (870 mAhg^−1^) and is easy to prepare. However, repeated lithium will lead to structural damage and poor cyclic stability of the material. Wang, et al. firstly prepared CoS_2_/C composites using a self-mounted template method, whose reversible capacity was about 700 mAh g^−1^ for the first time and specific capacity was 560 mAh g^−1^ after 50 cycles [10]. It can be concluded that reducing metal sulfide particles to the nanometer scale can significantly increase the specific surface area of the material and shorten the ion diffusion path, thus alleviating the volume effect of the material [11]. This study used the hydrothermal approach for efficiently synthesizing the 3D CSG nanocomposites, using C_3_H_7_NO_2_S as sulfur source and Co(NO_3_)_2_ ∙ 6H_2_O as cobalt source. The experimental results show that CSG composites have high structural stability, electronic, and ionic conductivity. Therefore, the specific capacity of the electrode material for the first discharge was 1176.1 mAhg^−1^ and the capacity retention rate was 61.5% after 100 cycles. When the charge-discharge were restored from 5.0 C to 0.2 C, the charge-discharge ratio could almost be restored to the initial capacity. In situ XRD was used to study the electrochemical transformation mechanism of the material. It was considered that the following advantages are largely responsible for its excellent performances, the unique micro and nano structure of the composite guarantees the stability of the electrode during cycling. The CoS_2_ nanoparticles embedded in rGO (with low density and large surface area) not only shortening the transmission distance between the ions and electrons but also slowing down the volume expansion during the charging and discharging process. The synergistic effect of nanoparticles and graphene improves the stability of electrical materials [12,13].

## 2. Experimental Section

### 2.1. Preparation of Materials

The detail information of chemical reagent in this experiment were shown in Appendix A. After referring to the classic literature on this subject, graphene oxide was prepared by an improved version of the graphene modified Hummer method [1,6]. The specific synthesis method was as follows: 3.0 g flake graphite was added into a beaker containing 60 mL concentrated H_2_SO_4_ (98%), and 1.5 g KNO_3_ was added under stirring condition. 9.0 g KMnO_4_ was added slowly after stirring in an ice water bath for 30 min. This process requires the temperature to be controlled at about 18 °C for 6 h. Then, it was transferred to 35 °C water bath for stirring for 24 h, slowly adding 10 mL H_2_O_2_ and 100 mL H_2_O. The mixture was centrifuged three times and placed into vacuum drying at 55 °C for 36 h. It was then ground to obtain pale yellow solid powder (GO). 30 mL of GO (1.0 mg/mL) was magnetic stirred for 30 min and then ultrasonicated for 2.5 h, followed by adding a solution (30 mL) of CoCl_2_ ∙ 6H_2_O (16 mg/mL). In order to complete ion exchange, the solution will continue to be stirred for another 3h to form solution A. The solution of L-cysteine (25 mg/mL) was added dropwise into solution (A) with stirring for 2 h. Aqueous solution (5 mL) of NaBH_4_ (5 mg/mL) was dropwise into above suspension (A) under stirring, the dark brown solution was transferred into a 100 mL Teflon-lined autoclave and heated at 120 °C for 18 h. After cooling down to room temperature, the resulted black solid was collected and was washed three times with deionized water and ethanol, respectively. After dried at 60 °C for 6 h, CoS_2_@rGO (CSG) was obtained. The control sample of pure CoS_2_ was prepared without the addition of GO powder. But all the other conditions and step remained the same. The CSG morphology obtained by the amount of GO and CoCl_2_ ∙ 6H_2_O is shown in Appendix A.

### 2.2. Characterization

The morphologies and texture of the samples were examined by field emission scanning electron microscopy (FE-SEM, Hitachi S-4800, Japan Electronics Corporation, Tokyo, Japan), and the element distribution was investigated with an energy-dispersive spectroscopy (EDS) detector (HITACHI, Tokyo, Japan). The crystal structures of the CoS_2_@rGO nanocomposite and pure CoS_2_ sample were studied by a X’ Pert3 Powder X-ray diffractometer (XRD, Tube pressure 40 kV, tube flow 20 mA, Cu target Kα ray, Panalytical B.V) at room temperature. In situ X-ray diffraction (Malvern Panalytical, Sahnghai, China) test conditions: Cu target Kα source, tube voltage 45 KV, tube current 40 mA, scan range 10–90°); X-ray photoelectron spectroscopy (XPS) was carried out on an ESCALab 250Xi spectrometer (Thermo, Waltham, MA, USA) with an Al Kα source and the C 1s peak as the internal standard at 284.8 eV. Raman spectra were con-ducted on an Invia Micro-Raman spectrometer (Renishaw, London, UK) with an excitation wavelength of 532 nm. Vacuum glove box (Braun Inert Gas Systems Co., LTD, Garching, Germany); constant temperature vacuum drying oven (Dongguan Bell Test Equipment Co., LTD, Dongguan, China); and thermal gravimetric analysis (TGA) was carried out by a Mettler Toledo TGA/SDTA 851 thermal analyzer under an air atmosphere. Nitrogen adsorption/desorption isotherms were obtained using a MicromeriticsASAP-2020HD88 analyzer at 77 K, and the pore size distribution curves were derived from the density functional theory method. Electrochemical measurements were performed on a LAND-CT2001A battery testing system (Wuhan, China). Cyclic voltammograms (CV) measurements were conducted on an electrochemistry workstation (CHI660B, Shanghai, China). The electro-chemical impedance spectroscopy (EIS) measurements also were performed over a frequency range from 0.1^−1^ Hz to 10^−5^ Hz and a signal amplitude of 10mV.

### 2.3. Fabrication of the Half Cells

Detailed information on the reagents, fabrication of the Half Cells is given in the Appendix A.

## 3. Results and Discussion

Figure 1 illustrates the preparation procedure of a 3D CSG composites. Firstly, the graphene oxide was prepared by an improved version of the graphene modified Hummer method. The surface of GO sheets is negatively charged due to the ionization of epoxyl, carboxyl and hydroxyl groups on it [14]. Co^2+^ ions formed a cobalt ion thiolate precipitate in the presence of L-cysteine in aqueous suspension of GO. In hydrothermal conditions, cobalt ion thiolate was decomposed and formed CoS_2_ NPs. In the meantime, GO were reduced to rGO by NaBH_4_. The CSG sample was thus achieved.

### 3.1. Morphology Analysis

Figure 2a,b shows the TEM of CSG composite. The CoS_2_ nanoparticles are anchored on the surface of multilayer rGO. As can be seen from Figure 2c,d, the particle size of CoS_2_ nanoparticles is uniform and concentrated in the range of 30–50 nm. Figure 2e is the high-resolution TEM image of the CSG composites. A uniform crystal lattice with the d-spacing of 0.256 nm matches well with the (111) plane of CoS_2_ [15]. Figure 2f shows the selected electron diffraction image of CSG composite. The diffraction period and polycrystalline diffraction point indicate that the composite have a polycrystalline structure, corresponding to the crystal planes of (200), (210), (211), (220), and (311) of CoS_2_ nanoparticles [12,16]. Figure 2g–j is the elemental mapping images of CoS_2_@rGO composite materials, respectively, which demonstrates the uniform elemental distribution of Co, S, and C in both samples. It illustrates the uniform distribution of CoS_2_ particles on rGO laminates further. The flexible reduction graphene oxide with large specific surface area can not only form a 3D conductive network structure, shorten the diffusion distance of lithium ions, and improve the electrical conductivity of the materials, but also buffer the volume expansion of CoS_2_ particles in the process of charge and discharge, inhibit the pulverization and agglomeration of CoS_2_ particles, and improve the cyclic stability of the material [13,17].

### 3.2. Structure Analysis

Figure 3a was the XRD pattern of CoS_2_@rGO composites, pure CoS_2_ and rGO. The CSG composite also displays distinct characteristic peaks at 32.5°, 36.4°, 40.0°, 46.5°, 55.3°, 60.4° and 76.8°, all of those peaks well correspond to the (200), (210), (211), (220), (311), (230), (311), (230), and (420) lattice planes of typical CoS_2_ (JCPDS NO. 03-0772), respectively. And that corresponds to the single CoS_2_ characteristic peak. It indicates that the target substance of synthesis is polycrystalline pure phase of nano CoS_2_, and also indicates that the composite reduction graphene does not change the crystal structure of CoS_2_ nanoparticles. There is an obvious peak at 26.5° attributed to rGO, indicating that GO has been reduced to rGO [12,18]. Figure 3b was the Raman spectra of CoS_2_@rGO and GO. The peak values at 1354 and 1596 cm^−1^ should be the D-peak and G-peak of graphene, respectively. The intensity ratio of peak D to peak G of CSG (ID/IG = 1.27) is higher than that of the original GO (ID/IG = 1.14), indicating more defects after reduction [19]. The reason is that although the oxygen-containing functional groups between carbon layers are removed after hydrothermal reduction, some carbon atoms fall off together with them. The structure becomes more disordered and the degree of graphitization increases [20]. In the Raman spectrum of CSG, the 2D and D + G peaks become weak due to the isolation of CoS_2_ on rGO [21]. Figure 3c–e was a scanning electron microscope (SEM) image of rGO, pure CoS_2_ and CSG. It can be seen from the figure that CoS_2_ nanoparticles are attached to the surface of rGO, the rGO can prevent the agglomeration of active substances (Figure 3d,e). Appendix A shows that the morphological destruction is one of the main factors affecting capacity attenuation.

To further analyze the surface composition of CSG composition, we conducted full spectrum scan of XPS (Figure 4). Figure 4a shows that the composite contains Co, S, C and O elements and there are no other miscellaneous fronds. Part of O element comes from partially unreduced GO. The XPS spectrum of Co 2p is shown in Figure 4b. 778.6 eV and 781.6 eV, which belong to Co 2p_3/2_ of Co-S, The two peaks located at 793.7 eV and 797.3 eV, which belong to Co 2p_1/2_ of Co-S [22]. Figure 4c. is the XPS spectrum of S 2p, revealing three bond types of S. The two strong peaks located at 163.5 eV and 165.3 eV are attributed to S 2p_1/2_ and S 2p_3/2_, respectively, which are caused by the split of S 2p spinorbit in the C-S part [23,24], proved that the S atom has been embedded in the carbon layer The two weak peaks at 168.5 eV and 169.7 eV correspond to S after oxidation, which is caused by oxidation of oxygen in air. Figure 4d, shows the C1s spectrum of graphene composites and the peak at 284.3 eV corresponds to Sp^2^ carbon. In addition, the three peaks at 285.3, 286.4 and 289.6 eV correspond to C–S, C=O and O=C-O group, respectively [25].The intensity of C–O, C=O and O=C-O peaks were obviously weaker of C–C, indicating that rGO has a higher degree of reduction [26,27]. The positions and intensities of the four main peaks also directly prove the existence of rGO in the composites.

### 3.3. Electrochemical Analysis

Figure 5a displays the first three cyclic voltammetry curves (CV) of CSG composite materials in the voltage range of 0.1–3 V. In the first cycle, the reduction peak at 0.9 V. corresponds to the insertion of Li^+^, and the formation of the solid electrolyte interface (SEI) [28] The small peak at 0.7 V can be attributed to the conversion reaction of Li_2_S.[29] To be precise, the reduction peaks occurring at 0.9 and 0.7 V can be attributed to the reaction: (1) CoS_2_ + *x*Li^+^ + *x*e^−^ = Li*_x_*CoS_2_ (2) Li*_x_*CoS_2_ + (4−*x*)Li^+^ + (4−*x*)e^−^ = Co + 2Li_2_S [30,31]. A sharp peak appeared at 0.49 V, which was mainly due to the intercalation of graphitized carbon. The oxidation peaks at 1.8 and 2.01 V are related to the formation of metal sulfides. In the subsequent cycles, three reduction peaks were observed at 1.5 V, 0.9 V, and 0.49 V, indicating that the structure has undergone a clear change and improved kinetics after initialization. This difference can be attributed to micro-evolution. The three oxidation peaks correspond to the extraction of Li^+^ from the electrode, attributable to the formula: 2Li_2_S + Co = CoS_2_ t 4Li^+^ + 4e^−^ [32]. The peak of 2.0 V was split into two peaks of 1.8 and 2.01 V, indicating that some changes occurred in the process of Li^+^ removal after the first cycle scan. The electrochemical properties of anode materials were studied by constant current charge/discharge test. As shown in Figure 5b, two voltage platforms appear at 1.40–1.32 V and 0.80–0.75 V for the first discharge. During the first charge, two voltage platforms appear at 2.00–2.06 V and 2.30–2.40 V. These results are consistent with CV results. The CoS_2_@rGO electrode exhibits a high charge-discharge capacity of 786.2 mAhg^−1^ and 1101.6 mAhg^−1^, and the Coulomb efficiency of the first cycle is 71.03%. The loss of the first cycle capacity is caused by the side reaction between electrolyte and polysulfide and the formation of SEI [33]. Figure 5e, shows the cyclic properties of CSG composites at 0.2 C, 3.0 C and CoS_2_ at 0.2 C (1 C = 870 mAhg^−1^). It can be observed from the figure that after the first charge-discharge cycle of CSG at 0.2 C, the capacity attenuates significantly due to the formation of solid electrolyte interface film (SEI) and side reactions in the charge-discharge process [29,34]. The coulombic efficiency of the first charge and discharge is only 73.8%. From the fifth charge-discharge cycle, the Coulomb efficiency is almost 100%, and the electrode material shows good cycling performance. After 100 cycles, the reversible capacity was 739 mAhg^−1^. The capacity also attenuated after the first charge-discharge cycle at 3.0 C, and the coulombic efficiency of the first charge-discharge cycle was only 69.2%. Start with the fifth charge and discharge cycle. The electrode material showed good cycling performance. After 305 cycles, the reversible specific capacity is 288 mAhg^−1^. The specific discharge capacity of pure CoS_2_ -was 1035.5 mAhg^−1^ after the first cycle and down to 112.1 mAhg^−1^ after the 200th cycles. This shows that the introduction of rGO improves the structural stability of the composites electrode, thus improving the first coulomb efficiency, reversible capacity and electrode cycle stability [32,35]. The rate performance of CSG and pure CoS_2_ nanomaterials -was shown in Figure 5c. The discharge capacity of CSG and pure CoS_2_ nanomaterials was similar in the first two cycles at a low rate of 0.2 C. When the rate increases gradually, the discharge capacity of CSG materials was significantly higher than CoS_2_ nanomaterials without rGO. The reversible capacities of CSG electrode materials are 836 mAhg^−1^, 724 mAhg^−1^, 684 mAhg^−1^, 611 mAhg^−1^, and 416 mAhg^−1^ at the current density 0.2 C, 0.5 C, 1.0 C, 2.0 C, and 5.0 C, respectively. The current density returns to 0.1 C, the CSG anode rapidly returns to a high capacity of 788 mAhg^−1^. These phenomena indicate that the CSG has excellent rate capability and structural integrity [36]. In addition, the specific discharge capacity of CSG composite is significantly higher than pure CoS_2_ nanomaterials when the ratio is recovered from 5.0 C to 0.2 C, and the reversible capacity of the composites electrode can be recovered to 713 mAhg^−1^. The results shows that CoS_2_ nanomaterials composite rGO was beneficial to improve the rate performance of the material. In order to study the electrochemical reaction kinetics of CSG electrode, the AC impedance between CSG and pure CoS_2_ electrode was measured. Figure 5d is original and fitted AC impedance spectra after the first charge and discharge by using Zview software (Version: 3.3c, Solartron, Scribner Associates Inc., Southern Pines, NC, USA). Compared with pure CoS_2_ nano material, the electrochemical impedance spectroscopy (EIS) of CSG composites showed a smaller half-cycle curve and a lower straight slope. CSG has an ohmic resistance (RW) of 3.86 ω, close to CoS_2_ (3.78 ω). CSG has a charge transfer resistance (Rct) value of about 67.3 ω, which is very low compared to CoS_2_ (about 282.5 ω) [36,37]. This is due to the fact that the transfer rate of lithium ions in CSG composites is higher than that of the pure CoS_2_, which improves the electrochemical properties of the composites [38].

The excellent electrochemical performance of CSG is related to the following factors [16,20,39]: (1) the small size of CoS_2_ nanoparticles shortens the diffusion path of lithium ions and provides more electrochemical active sites; and (2) the introduction of flexible large specific surface area reduction graphene not only provides 3D conductive network structure for composite materials, but also improves the electrical conductivity of the material. It was advantageous to the electrolyte seepage, promoted the rapid transmission of lithium ion and the ability to buffer CoS_2_ volume expansion in the process of charging and discharging, contributed to the suppression of CoS_2_ nanoparticles pulverization and reunion, and improved the cycle stability of the material.

Figure 6c reveals the lithium storing schematic of CoS_2_. The first step is to convert cobalt and lithium sulfide to cobalt and lithium sulfide. This step is usually an irreversible reaction. The second step is to form LiCoS_2_. This step is the reversible step for lithium storage [34,38]. To study the relationship between the electrochemical activity of the electrode material of CSG and the structural changes of the active material, we tested the in-situ XRD of the first cycle of constant-current charge and discharge of the half-cell (in Figure 6a and Appendix A). Figure 6b was the initial charge-discharge curve. The XRD patterns show that there are five strong peaks at 46.5°, 50.6°, 52.8°, 71.4°, and 85.0°, which belong to the characteristic peaks of Be (JCPDS No. 02-1366). For the original electrode, the peak of the XRD pattern is CoS_2_. When the battery is discharged to 1.4 V, the peak value of CoS_2_ becomes weak, indicating that some reaction has taken place. When the electrode was further discharged to 0.9 V, the characteristic peaks of Li_2_S (JCPDS No. 77-2145) and Co (JCPDS No. 01-1277) could be detected, indicating that the conversion reaction occurred. XRD results shows that CoS_2_ is transformed into Li_2_S and Co without Li insertion at the initial lithium stage. In the further lithium process, the diffraction peak at 47.3° gradually increases, indicating the presence of more and more metallic Co. After further discharge, no peaks of Co and Li_2_S are detected, indicating that LiCoS_2_ has been completely transformed, and the structural phase of CoS_2_ is regulated by the formation of intermediate LiCoS_2_ [33,36]. The results shows that the continuous discharge below 0.9 V eventually leads to the conversion reaction of LiCoS_2_ to Li_2_S and Co. The electrochemical process is as follows: Li*_x_*CoS_2_ + (4 − *x*) e^−^ + (4 − *x*)Li^+^ = Co + 2Li_2_S. When fully charged, a weak and wide CoS_2_ diffraction peak is found at 27.8°, which can be attributed to the reaction process of Co + 2Li_2_S = CoS_2_ + 4Li^+^ + 4e^−^, indicating that the reaction has the good reversibility of an electrochemical reaction [28,40]. The electrochemical performance comparisons between the CSG electrode and some typical transition metal chalcogenides as anode materials for LIBs are shown in Appendix A. The result indicates that the CSG composites also has great application potentials as a high-rate anode material for LIBs.

## 4. Conclusions

In this work, CSG nanocomposites were synthesized by a hydrothermal method and characterized by in-situ XRD, TEM, HRTEM, and SEM. This resulted in the successful preparation of the composites and the uniform dispersion and anchoring of nanoscale CoS_2_ on the surface of rGO. The electrochemical performance of pure CoS_2_ electrode was compared with that of composite electrode, and the composite electrode showed excellent cycling and rate performance. The storage mechanism of lithium composites was investigated by in-situ XRD, and the conversion mechanism was explained. CoS_2_ nanoparticles were dispersed on the surface with two-dimensional conductive skeleton rGO, which can not only prevent the agglomeration of nanoparticles but also facilitate the rapid shuttle of electrons. It has a positive effect on the stability of electrode material structure. These results indicate that CSG was a good candidate as an electrode material for energy storage. This new strategy is simple and has a low cost. It may be applied to fabricate other micro/nano sulfide electrode materials for superior LIBs. The electrochemical properties of CSG make it a promising new LiBs anode material.

## Figures and Tables

**Figure 1 nanomaterials-12-00724-f001:**
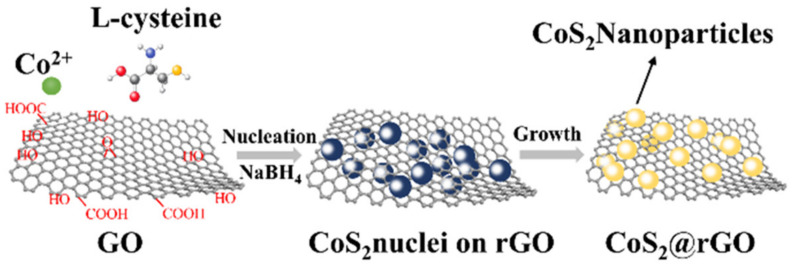
Scheme of the synthesis route of CoS_2_@rGO.

**Figure 2 nanomaterials-12-00724-f002:**
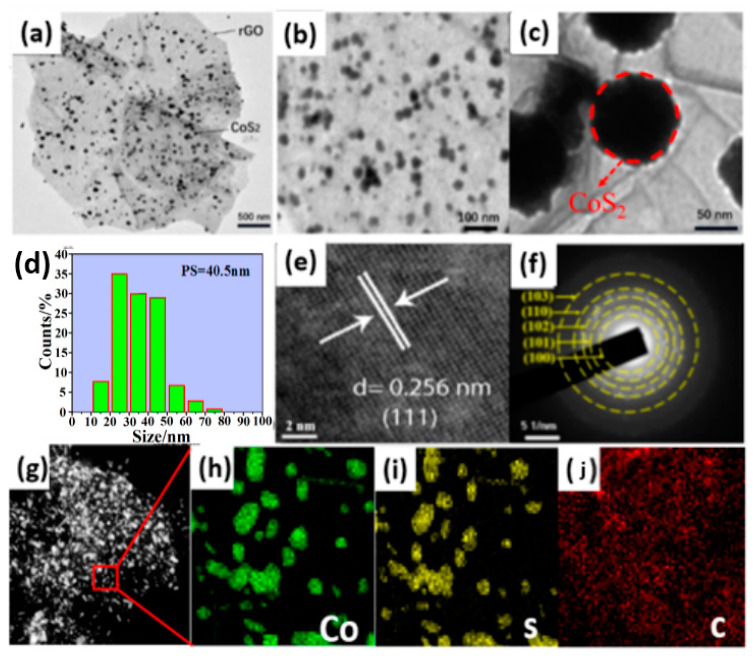
(**a**–**c**) TEM images of CoS_2_@rGO. (**d**) The particle size distribution of CoS_2_@rGO composites. (**e**,**f**) HRTEM image and selected area electron diffraction pattern of CoS_2_@rGO. (**g**–**j**) Elemental mapping images of CoS_2_@rGO composites.

**Figure 3 nanomaterials-12-00724-f003:**
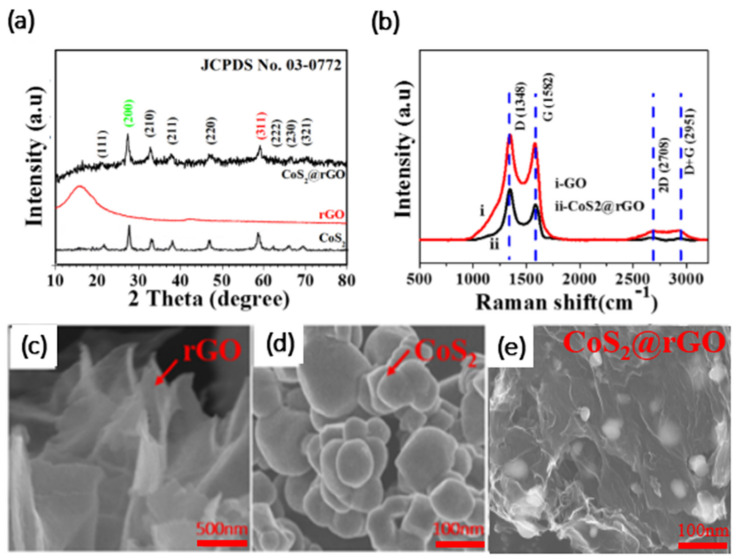
(**a**) XRD pattern and (**b**) Raman spectra of CoS_2_@rGO; (**c**–**e**) SEM images of rGO, CoS_2_, CoS_2_@rGO.

**Figure 4 nanomaterials-12-00724-f004:**
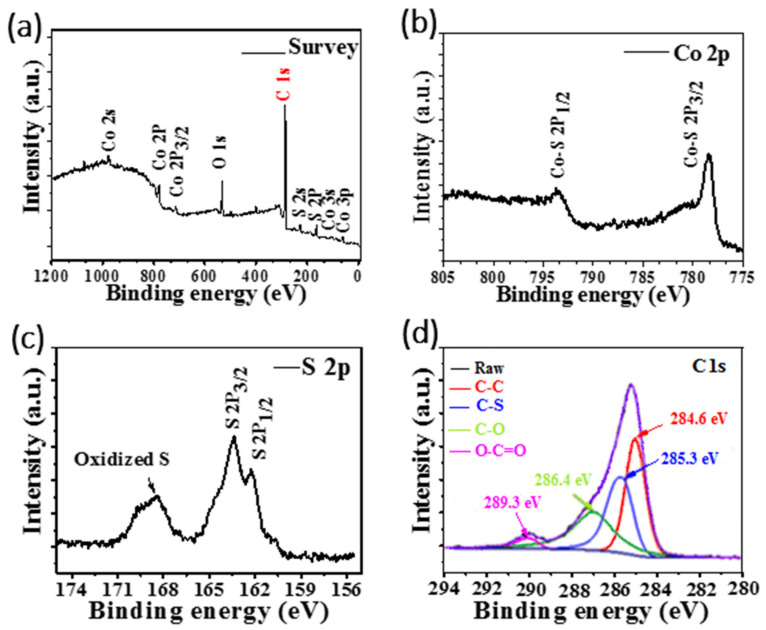
XPS spectrum of CSG composite: (**a**) survey spectrum, (**b**) Co 2p, (**c**) S 2p, and (**d**) C 1s of XPS spectrum of CoS_2_@rGO.

**Figure 5 nanomaterials-12-00724-f005:**
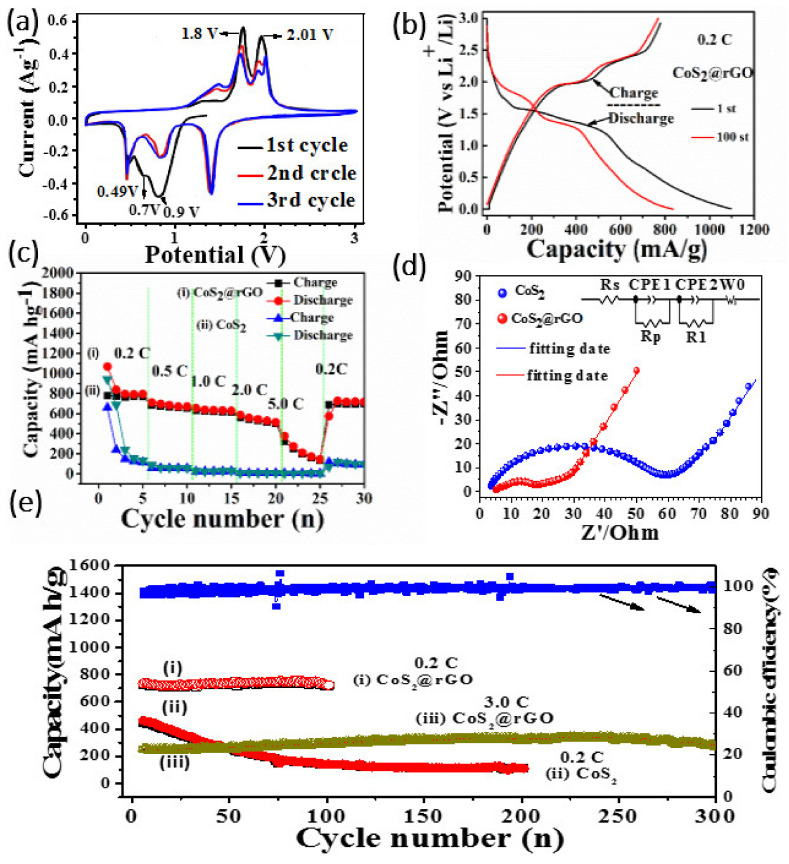
(**a**) CV curves of CSG nanocomposites in the voltage range of 0.1-3.0 V at 0.1 mV s^−1^, (**b**) Charge-discharge performances at various rates, (**c**) Rate performance of (i) CoS_2_@Rgo (ii) CoS_2_, (**d**) EIS spectra of CSG and CoS_2_, and (**e**) The cycle performance of CSG at 0.2 C (i), 3.0 C (iii) and cycle performance of CoS_2_ at 0.2 C (ii).

**Figure 6 nanomaterials-12-00724-f006:**
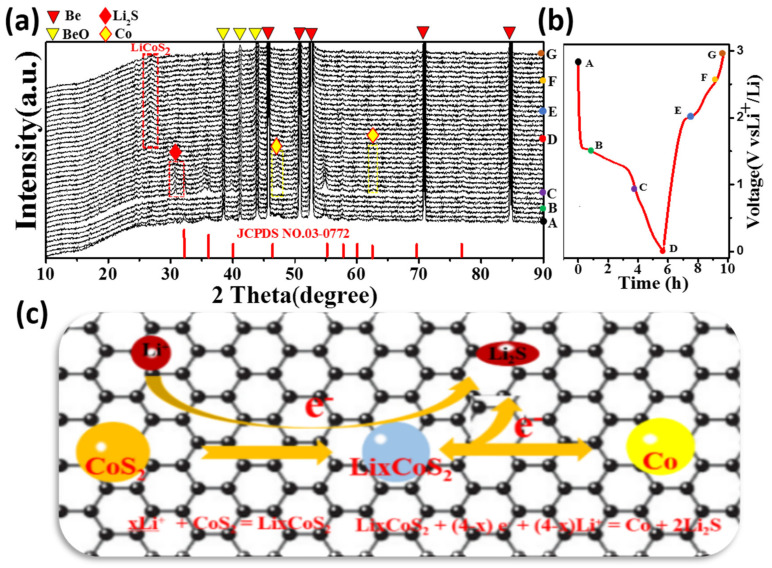
(**a**) In situ XRD investigation of CSG nanometers Cut-off: 0.001-3.0 V versus Li/Li^+^. Current density: 0.2 C. (A,B,C,D,E,F,G correspond to XRD at each voltage value); (**b**) The corresponding XRD pattern during the first cycle. (**c**) Schematic diagram of the mechanism of CSG storing lithium.

## Data Availability

Data presented in this article is available on request from the corresponding author.

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
