# Peer review of "Composite Nanoarchitectonics with CoS2 Nanoparticles Embedded in Graphene Sheets for an Anode for Lithium-Ion Batteries"

_nanomaterials, 2022, doi:10.3390/nano12040724_

Round 1
Reviewer 1 Report
The manuscript presents the preparation of a CoS2/rGO composite, and its use as anode in Li-ion batteries. Reported data are extensive, and generally its analysis and discussion meaningful. Main exception are the following: - In Fig. 5 peak attribution and equations provided are contradictory, in particular it is not clear what is SEI formation, and Li interaction with graphitized carbon or Co compounds. - In Fig. 6 the diffractograms do not show continuous evolution. Li2S peaks between C and D appear and then disappear suddenly; upon charge CoS2 appears again. The illustration of Fig.6c does not seem to explain this behavior. - The relevance and challenges of using sulfides as Li anode materials is not addressed in the introduction - The instruments used is provided as list, without details on conditions in the experimental part; it is not clear what has been the use of the Field emission electron microscope. The presentation is generally poor, graphics have low resolution and are distorted (e.g. diffraction rings are elliptic), XPS horizontal axis is plotted with increasing BE, while Nyquist diagram coordinates are not squared. Text is poor in formatting, spelling and grammar. I suggest careful revision and professional proofreading.Author Response
Please see the attachment

Reviewer 2 Report
The paper reported that synthesis and characterization of CoS2@rGO including the performance evaluation of a secondary butteries. The paper is well written and organized. I think that following modifications are required for publication.
- Page 2, line 69: Nano should replace to nano.
- Page 2 line 98 and Page 3 line 100: the "pressure in line 98 is the same as voltage in line 100. Please use the same word. The flow in line 98 is also the same as current in line 101. Please use one kind of word to avoid confusing.
- Page 3, line 114, ESI.. should change to ESI.
- Page 5, line 152: The counting statistics of the powder data of the CoS2@rGO is very low. Please remeasure the data with sufficient x-ray exposure time measurement. In the present data, the noise level is comparable to the peaks of CoS2@rGO
- Page 9, line 280: Please show an expanded region from 20 deg. to 50 deg. in two theta. It is very difficult to recognize the peaks. I think that the time dependence of peak intensities for CoS2 and Li2S is useful for the reader to understand the phenomena.
Round 2
Reviewer 1 Report
The authors answered satisfactorily to most points raised. However, XPS horizontal axis still need be reverted. In addition, graphics resolution and language, including spelling in the text (e.g. “charge storing” instead of “charge storage”) and graphics (e.g. title of Fig. 4a) remain unsatisfactory. I hope the Editorial Office can deal with that.
